# Tailor-made nanostructures bridging chaos and order for highly efficient white organic light-emitting diodes

Yungui Li[1], Milan Kovačič[2], Jasper Westphalen[3], Steffen Oswald[4], Zaifei Ma[5], Christian Hänisch[1], Paul-Anton Will[1], Lihui Jiang[1,6], Manuela Junghaehnel[3], Reinhard Scholz[1], Simone Lenk[1] & Sebastian Reineke[1]

Organic light-emitting diodes (OLEDs) suffer from notorious light trapping, resulting in only moderate external quantum efficiencies. Here, we report a facile, scalable, lithography-free method to generate controllable nanostructures with directional randomness and dimensional order, significantly boosting the efficiency of white OLEDs. Mechanical deformations form on the surface of poly(dimethylsiloxane) in response to compressive stress release, initialized by reactive ions etching with periodicity and depth distribution ranging from dozens of nanometers to micrometers. We demonstrate the possibility of independently tuning the average depth and the dominant periodicity. Integrating these nanostructures into a two-unit tandem white organic light-emitting diode, a maximum external quantum efficiency of 76.3% and a luminous efficacy of 95.7 lm W$^{-1}$ are achieved with extracted substrate modes. The enhancement factor of 1.53 ± 0.12 at 10,000 cd m$^{-2}$ is obtained. An optical model is built by considering the dipole orientation, emitting wavelength, and the dipole position on the sinusoidal nanotexture.

[1] Dresden Integrated Center for Applied Physics and Photonic Materials (IAPP) and Institute for Applied Physics, Technische Universität Dresden, Nöthnitzer Str. 61, 01062 Dresden, Germany. [2] University of Ljubljana, Faculty of Electrical Engineering, Tržaška 25, 1000 Ljubljana, Slovenia. [3] Fraunhofer Institute for Organic Electronics, Electron Beam and Plasma Technology FEP, Winterbergstraße 28, Dresden 01277, Germany. [4] Institute for Complex Materials, Leibniz IFW Dresden, Helmholtzstraße 20, Dresden 01069, Germany. [5] Center for Advanced Low-dimension Materials, State Key Laboratory for Modification of Chemical Fibers and Polymer Materials, Donghua University, 201620 Shanghai, China. [6] College of Chemistry and Chemical Engineering, Central South University, 410083 Changsha, China. Correspondence and requests for materials should be addressed to S.R. (email: sebastian.reineke@tu-dresden.de)

Organic light-emitting diodes (OLEDs) have gained tremendous attention from both academic and industrial communities for more than three decades. Within this timeframe, the efficiency, reliability, and brightness have dramatically improved to a level suitable for commercial display applications. The focus has therefore moved towards improving the performance of white OLEDs for lighting applications, a field that poses much stricter requirements including stability, stable angular emission, and comparable power efficiency to the currently used techniques. Since the first realization of white OLEDs, great efforts have been made to achieve a balanced white spectrum and high luminous efficacy at a practical luminance level[1–4]. With the development of phosphorescent[5,6] and thermally activated delayed fluorescent (TADF) emitters both able to fully harvest triplet excitons in devices[7,8], doped transport layers[9], and efficient blocking architectures[2], the internal quantum efficiency of white OLEDs can now reach 100%[2,4]. However, the external quantum efficiency (EQE) for devices without additional outcoupling techniques can only reach 20–40%. About 20% of the generated photons are trapped inside the glass substrate due to the total internal reflection (TIR) at the glass and air interface as substrate modes. Another proportion (40–60%) of photons is waveguided in the organic layers ($n_{org} \approx 1.7$) and indium tin oxide (ITO, $n_{ITO} \approx 1.8$) because of the lower refractive index of the glass substrate ($n_{sub} \approx 1.5$). The third part of optical losses is due to trapped photons (20–40%) at the interface between the organic layer and the top metal electrode as surface plasmon polariton (SPP) modes[10].

Numerous approaches have been investigated to extract the trapped photons from OLEDs. Regarding the total internal reflection losses at the substrate-air interface, methods such as modifying the substrate surface with a microlens array[11], hierarchical ultrastructures from fireflies[12], and scattering nanoparticles[13,14] have been introduced. These approaches can only extract light from substrate modes but neither from waveguide modes nor SPP modes. Concepts such as using high refractive index substrates[3], low-index grids between anode ITO and organic layers[15], and sub-anode grids between substrate and ITO layer[16] have been demonstrated to extract waveguide modes, but these techniques have drawbacks either including toxic components, showing angular or wavelength-dependent enhancement behaviors, or involving complicated lithography processes. Meanwhile, to extract photons trapped as SPP modes, periodic structures such as optical gratings[17] and photonic crystals[18], have been introduced for monochromatic OLEDs. However, the angular or wavelength-dependent emission hinders the application for lighting purposes. Previously reported nanostructures such as bucklings[19] and bioinspired nanostructures[20] showed impressive enhancement on device efficiency. However, complicated processes such as multiple heating, lithography, molding, and/or nanoimprinting steps are involved, making them rather unsuitable for large-scale and low-cost production. Facile and controllable techniques are still needed.

Here, we report a method to extract trapped photons in white OLEDs by using controllable quasi-periodic nanostructures with a broad periodicity and depth distribution from tens of nanometers to micrometer range, which are induced by reactive ions etching (RIE) on poly(dimethylsiloxane) (PDMS) surfaces for a short time. The average depth and periodicity distribution can be controlled by tuning the conditions of the pretreatment of PDMS and the RIE processing parameters. The dominant periodicity and average depth can be controlled simultaneously or independently. To estimate the influence of the nanostructure on the power dissipation of white OLEDs, we propose an optical model to numerically simulate the dissipated energy to the substrate medium, by treating the nanostructure as a sinusoidal nanotexture. Considering parameters including the size of the nanostructure, the emission wavelength, the dipole orientation, and the dipole position on the nanostructure, the simulated enhancement factors are very close to our experimental results. By applying these nanostructures to ITO-based two-unit tandem white OLEDs, the air mode ($\eta_A$, external quantum efficiency without external outcoupling technique) and the total quantum efficiency of substrate modes and air modes ($\eta_{SA}$, external quantum efficiency with a glass hemisphere attached to the substrate) are significantly enhanced. It is possible to enhance $\eta_{SA}$ by a factor of $1.53 \pm 0.12$ at $10,000 \, \text{cd m}^{-2}$, without introducing angular or wavelength-dependent emission. In total, the $\eta_{SA}$ can reach up to 76.3% and the luminous efficacy up to $95.7 \, \text{lm W}^{-1}$. The controllable generation of these nanostructures proves to be facile and lithography-free. Therefore, it could be a promising outcoupling technique for large area lighting applications.

## Results

**Nanostructure generation and characterization.** The generation of the nanostructures is schematically illustrated in Fig. 1a. Undulations with dimensions from nanometer to micrometer scale are found on the entire surface of the PDMS after a short duration of RIE treatment with oxygen and/or argon flow, as shown in Fig. 1b. The appearance of these patterns is similar to mechanical instabilities such as creases, folding, or wrinkles induced by non-equilibrium on multilayered surfaces[21–24]. Cross-section profiles from the atomic force microscopy (AFM) measurement with sinusoidal shape can be found, with amplitudes ranging from tens to hundreds of nanometers. The ring shape of the Fast Fourier Transform (FFT) pattern in Fig. 1c indicates a random distribution in all directions. The calculation of the radial power spectral density function (PSDF) from AFM measurements shows a widely distributed periodicity from less than 100 nm to more than 1000 nm with a dominant periodicity for each individual nanostructure, as depicted in Fig. 1c. There is also a depth distribution with a dominant depth $p$ for these nanostructures (Supplementary Fig. 1). The average depth of sinusoidal structure can be described as $D = 2R_a$, where $R_a$ refers to the average roughness obtained from the AFM measurement[25]. The detailed analysis of different depth parameters can be found in Supplementary Note 4 and Supplementary Fig. 13.

The experimental repeatability is monitored by measuring the periodicity and depth for the nanostructure generated in different batches with the same recipe as a tracking sample. As shown in Supplementary Fig. 3, in multiple batches fabricated at different time, the deviation of the dominant periodicity and the average depth of the tracking sample is very small, demonstrating that the method is controllable with good experimental repeatability. Similarly, the uniformity of the nanostructures generated on the PDMS is checked across a large, macroscopic surface area. AFM measurements on different positions randomly chosen for each sample are carried out to locally probe the nanostructure parameters, i.e., periodicity and depth. As shown in Supplementary Fig. 2, the periodicity distribution of a specific structure (N1) is almost the same for all these measurements, while the deviation of the proposed aspect ratio (AR, AR = depth/periodicity) is very small, summarized in Supplementary Table 1 and Table 2, indicating the uniformity of the nanostructures across the entire surface.

It is possible to maintain the shape of the periodicity distribution without significant shift of the dominant periodicity while tuning the average depth by varying the RIE treatment power and duration time. As shown in Fig. 2a, the dominant periodicity is located at ~350 nm. There is only a slightly different distribution in the large periodicity range, when the RIE power is increased from 20 to 200 W, while the PDMS preparation is the same for each sample and

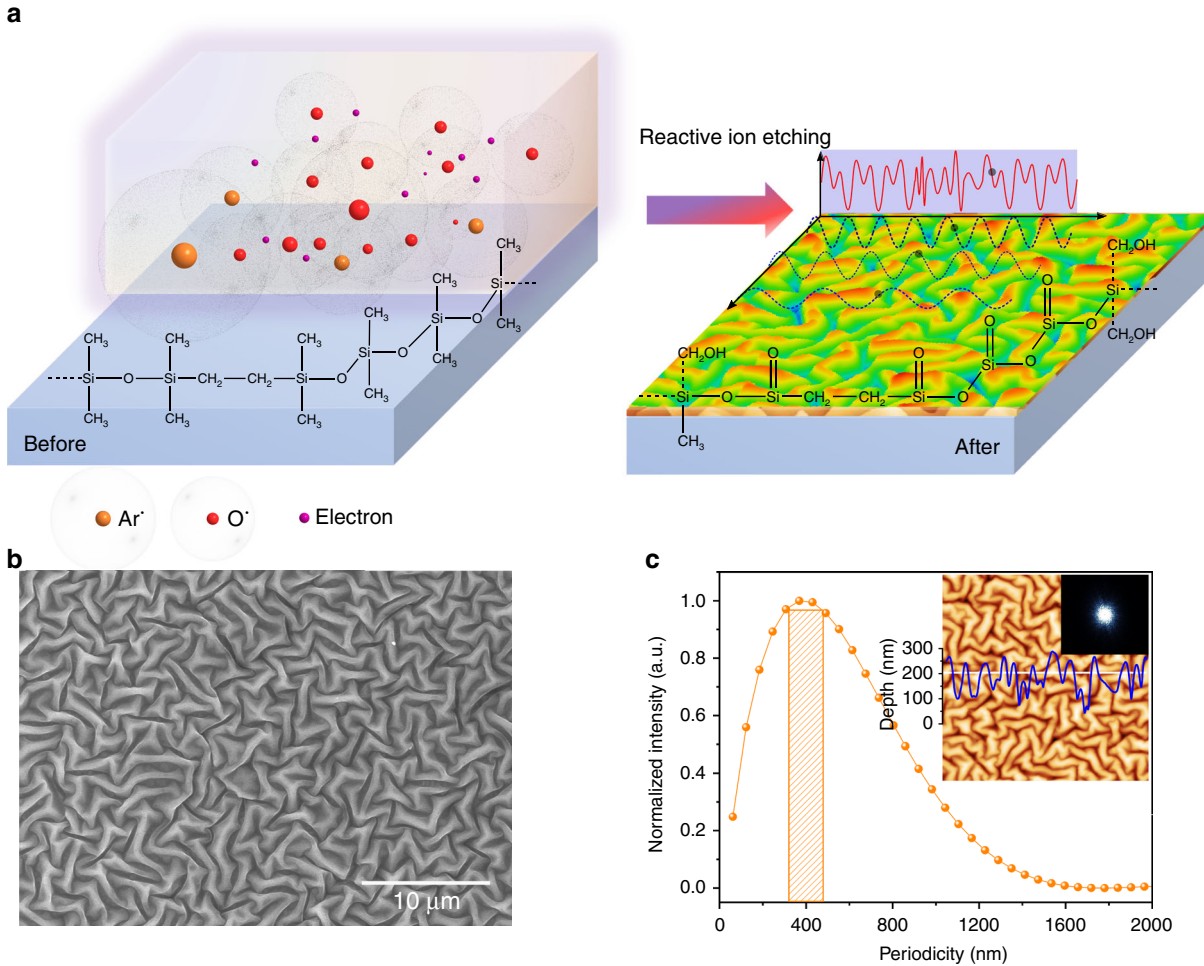

**Fig. 1** Schematic illustrations of RIE-induced nanostructures. **a** Thermally pretreated PDMS is modified by RIE with Ar, $O_2$ ions and electrons via physical bombardments and chemical reactions. After the RIE treatment, the surface of PDMS is turned to a silica-like layer with a topography of sinusoidal undulations. The presented chemical configuration of PDMS after RIE treatment is one of the possible structures. **b** SEM image. The nanostructure is widely and homogeneously distributed on the entire surface. **c** The periodicity distribution of the RIE-induced nanostructure. The orange bar indicates the dominant periodicity. Inset: AFM image with a size of 10 μm × 10 μm. The white line is placed at a randomly chosen position with a wavy profile. The ring shape of the inset FFT pattern indicates the nanostructure is oriented without directional preference. The nanostructures can be simplified to a sum of sinusoidal functions with different periodicities and heights

RIE treatment time is kept constant at $t = 60$ s. However, the average depth grows almost linearly from 20 to 120 nm (Fig. 2b).

As shown in Supplementary Fig. 4, a similar linear increase of the average depth is observed when extending the RIE treatment time, while maintaining the dominant periodicity. It is also possible to tune the periodicity distribution and the average depth simultaneously. Further investigations reveal that the change of the pretreatment time of the PDMS (Fig. 2c, d), the weight ratio of the base to the curing agent of PDMS (Fig. 2e, f), and the gas species and gas flow (Supplementary Fig. 4) can tune the periodicity and average depth at the same time. The detailed analysis between the fabrication data set and the dimensional parameters can be found in Supplementary Note 1. From all these observations, we conclude that it is possible to tune the average depth from ~10 to ~140 nm and the dominant periodicity from ~200 nm to ~800 nm, simultaneously or independently. We anticipate that the characteristic nanostructure parameters can be extended beyond the explored limits with a broader variation of those parameters.

**Mechanism of the nanostructure control**. As a next step, we explore the mechanism behind the nanostructure generation and controllability. To detect the chemical composition of the PDMS surface after the RIE treatment, an X-ray photoelectron spectroscopy (XPS) measurement is conducted and the result is shown in Fig. 3a. For the as-prepared PDMS before the RIE treatment, the Si 2p binding energy is 102.5 eV, which is in agreement with the previously reported value for PDMS[26]. After the RIE treatment, the Si 2p peak shifts to 103.1 eV. Here, the XPS spectra can be fitted with three different components, representing possible chemical structures (Si–O binding) of the surface layer. According to previous reports, the peak at ~102.2 eV represents the chemical configuration of $[(CH_3)_2SiO_{2/2}]$, ~103.2 eV $[(CH_3)SiO_{3/2}]$ and ~104.0 eV $[SiO_{4/2}]$[26–28]. For the as-prepared PDMS, the majority component is $[(CH_3)_2SiO_{2/2}]$, which is consistent with chemical configuration of PDMS. After the RIE treatment, the ratio of $[(CH_3)_2SiO_{2/2}]$ decreases, while the proportion of $[(CH_3)SiO_{3/2}]$ and $[SiO_{4/2}]$ increases. The atomic concentration (at%) of the as-prepared and RIE-treated PDMS is summarized in Supplementary Table 3. The as-prepared PDMS at the surface is composed of 27.8 at% oxygen and 46.2 at% carbon. However, the concentration of oxygen increases to more than 40 at% and the concentration of carbon decreases to less than 30 at% after the RIE treatment. The atomic concentration of silicon is almost the same after the RIE treatment. From these observations, it is

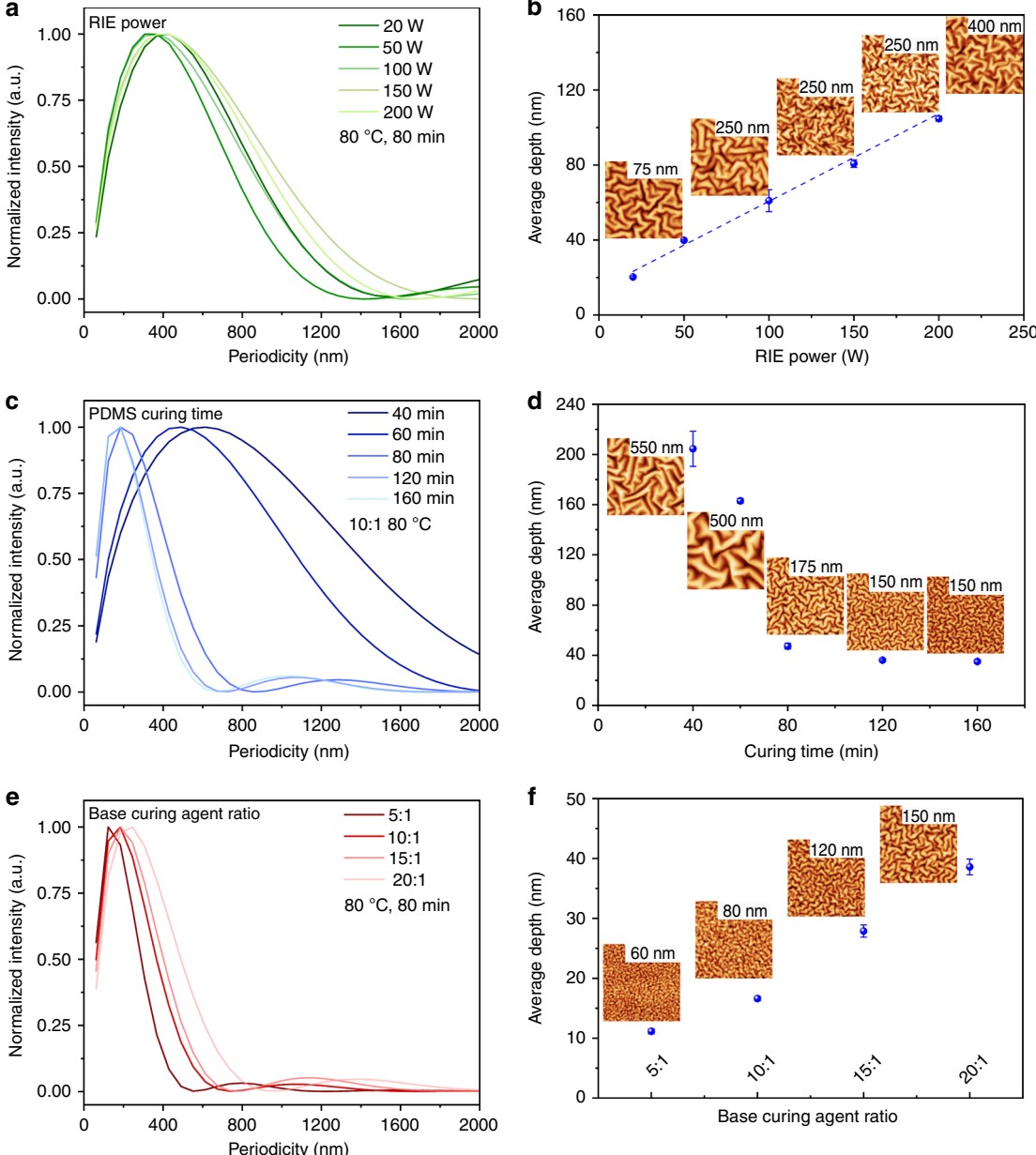

**Fig. 2** The control of RIE-induced nanostructures. The influence of the RIE power (**a**, **b**), the PDMS pretreatment time (**c**, **d**) and the weight ratio of the base and curing agent (**e**, **f**) on the periodicity distribution (**a**, **c**, **e**) and the average depth (**b**, **d**, **f**). The inset AFM figures are shown with a size of 5 μm × 5 μm, with different color scale from 0 nm to the maximum height marked for each figure. The detailed conditions for PDMS preparation and RIE treatment can be found in the method part. Error bar: the standard deviation from at least 3 AFM measurements on different sites

reasonable to deduce that the surface of PDMS is turned to a silica-like layer.

Identical XPS spectra are obtained for samples being treated with different RIE recipes, including various values of treatment power or gas flow rate, as summarized in Supplementary Table 4. The composition of the surface layer is independent from the treatment power when only oxygen is used as treatment gas. Moreover, different gas flow rates result in identical XPS spectra, indicating the composition of the top silica-layer does not change with different gas flows. This should be attributed to the fact that the silica-like layer on the PDMS generated by the RIE treatment hinders further treatment of the deeper layer of PDMS[28], and thus, gives the composition of the top layer of RIE-treated PDMS identical in the first several or dozens of nanometers. However, the real thickness of this silica-like top layer cannot directly be

measured, because of the wavy geometry and the strong adhesion to the soft PDMS base without a clear boundary.

To investigate the change of the modulus of the as-prepared PDMS with different heating time and weight ratio, the dynamic mechanical analysis (DMA) is done. Here, the storage modulus is regarded as Young's modulus in the stressing measurement mode[29]. As depicted in Fig. 3b, the extension of pretreatment time can increase the modulus, whenever the weight ratio of base to the curing agent is the same. It is 0.8 MPa for PDMS cured for 40 min and further increases to 1.6 MPa when extending the curing time to 160 min, while keeping the weight ratio at 10:1. The modulus also raises when the weight of curing agent in the mixture is increased. It changes from about 0.3 MPa to 1.6 MPa, when the weight ratio of the base to the curing agent varies from 20:1 to 5:1. This result can be ascribed to the fact that extending

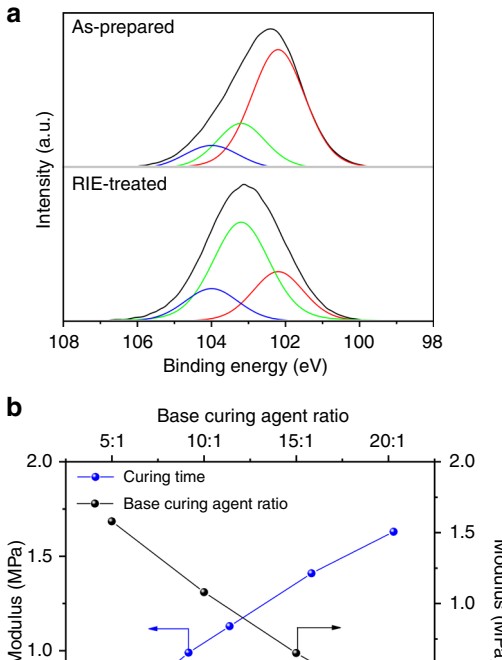

**Fig. 3** Mechanism investigation on the control of RIE-induced nanostructures. **a** The binding energy of the Si 2p peak and fitted components. Top: as-prepared PDMS. Bottom, the RIE-treated PDMS. **b** The storage modulus of PDMS after the thermal pretreatment. The pretreatment condition for PDMS with different weight ratios of the base to curing agent: 80 °C, 80 min. The thermal pretreatment condition for PDMS with a 10:1 weight ratio: 80 °C and varied time

the curing time or increasing the weight ratio of curing agent of PDMS leads to a higher cross-link level, which gives rise to a higher modulus.

Since the chemical composition of the silica-like layer is between the intrinsic PDMS and silica, it is reasonable to deduce that the Young's modulus of this layer ranges from MPa to GPa (modulus of intrinsic silica), which should be higher compared to the as-prepared PDMS and lower than that for pure silica. However, the exact value of the Young's modulus of the top silica-like layer cannot be measured directly due to the influence of the bottom soft PDMS[30]. Nevertheless, the silica-like layer generated at the surface can be regarded as a stiff layer in this bilayer system.

Based on these results, the generation and controllability of RIE-induced nanostructures can be explained by the theory of thin-film deformation in a planar bilayer system. Layered thin materials generate mechanical deformations such as creases, wrinkles and undulations on the surface in response to small compressive stress release between the top stiff layer and the bottom soft base, induced by thermal, light, mechanical, or osmotic stimuli[21,23,31–35]. Deformations arrange randomly in directions and amplitude, accompanied with in-plane stress release on the entire surface.

Oxygen or argon flow can turn to be reactive species such as radicals, ions and electrons by the interaction between the glow discharge and undissociated gases during RIE treatment[36]. These highly reactive species can modify the PDMS surface by chemical reactions and physical bombardments. The components of PDMS can be oxidized to volatile gases, which are removed by the

vacuum pumping system during the RIE treatment, transforming the RIE-treated surface into a form of silica-like composition. Deformation starts to minimize the total energy of the bilayer system, when the compressive stress $\sigma$ exceeds the critical level $\sigma_{crit}$, induced by the stimuli of RIE treatment and the modulus mismatch between the top stiff silica-like layer and the bottom soft PDMS. According to the thin-film deformation theory on the flat surface, the threshold $\sigma_{crit}$ is defined by the mechanical properties of the bilayer system[37–39]:

$$\varepsilon_{crit} = 0.52 \left[\frac{E_s}{(1-v_s^2)}\right]^{1/3} \left[\frac{E_{PDMS}}{(1-v_{PDMS}^2)}\right]^{2/3} \quad (1)$$

where subscript s indicates the stiff layer, $E$ Young's modulus and $v$ the Poisson's ratio.

The dominant periodicity $p$ and the average depth $D$ of the resulting sinusoidal pattern is given by:

$$p = 2\pi t_s \left[\frac{E_s(1-v_{PDMS}^2)}{3E_{PDMS}(1-v_s^2)}\right]^{1/3} \quad (2)$$

$$D = t_s \left(\frac{\sigma}{\sigma_{crit}} - 1\right)^{1/2} \quad (3)$$

where $t$ is the layer thickness.

When the compressive stress $\sigma$ is much larger than the critical stress $\sigma_{crit}$, the relation can be further simplified to:

$$D \sim t_s \sigma^{1/2} \quad (4)$$

The dominant periodicity $p$ can be influenced by $t_s$, $E_s$ and $E_{PDMS}$. According to Eq. 2, the dominant periodicity $p$ drops to a shorter range, when the modulus of PDMS ($E_{PDMS}$) is increased by extending the pretreatment time or increasing the weight ratio of the curing agent, which is confirmed by the DMA (see Fig. 3). In cases of the RIE treatment with only oxygen gas, increasing the RIE power or time has little impact on $E_{PDMS}$, $E_s$, or $t_s$, as revealed by XPS and DMA measurements. The dominant periodicity $p$ of the RIE-induced nanostructures can be maintained without pronounced shift. However, the depth of the nanostructures can be tuned by increasing the compressive stress through channels of physical bombardment and heat leakage from chemical oxidization. Similar linear relationships between the roughness of polymer surface and the plasma treatment power or time have been reported during plasma treatment of other polymer systems[40,41]. For RIE treatments with argon flow, it could be that the thickness of the top stiff layer $t_s$ and the compressive stress $\sigma$ are changed at the same time, leading to a variation of the periodicity distribution and depth simultaneously.

**Optical modeling of white OLEDs on RIE-induced nanostructures**. The application of quasi-periodic nanostructures in white OLEDs is investigated using optical modeling. First, planar white tandem devices are numerically evaluated using an in-house developed simulation tool (see Supplementary Note 2)[42]. The model previously showed a good agreement with experimental results for planar monochromatic OLEDs[43,44], as well as for tandem white devices[45].

Here, the quasi-periodic nanostructures can be simplified to a sum of sinusoidal functions with different periodicities and heights and their optical effect on OLEDs is evaluated using the finite element method (FEM)[46]. As shown in Fig. 4a, using 2D simulations, each single sine nanotexture with periodicity $p$ and height $h$ at the specific wavelength is modeled separately. The device configuration is following the one used in experiments. Optical modeling is done in the wavelength range from 400 to 800 nm with a step width of 10 nm. When focusing on extracting waveguide and SPP modes only, the outcoupling efficiency $\eta_{out}$ to the substrate is simulated. The simulated $\eta_{SA}$ can be obtained by

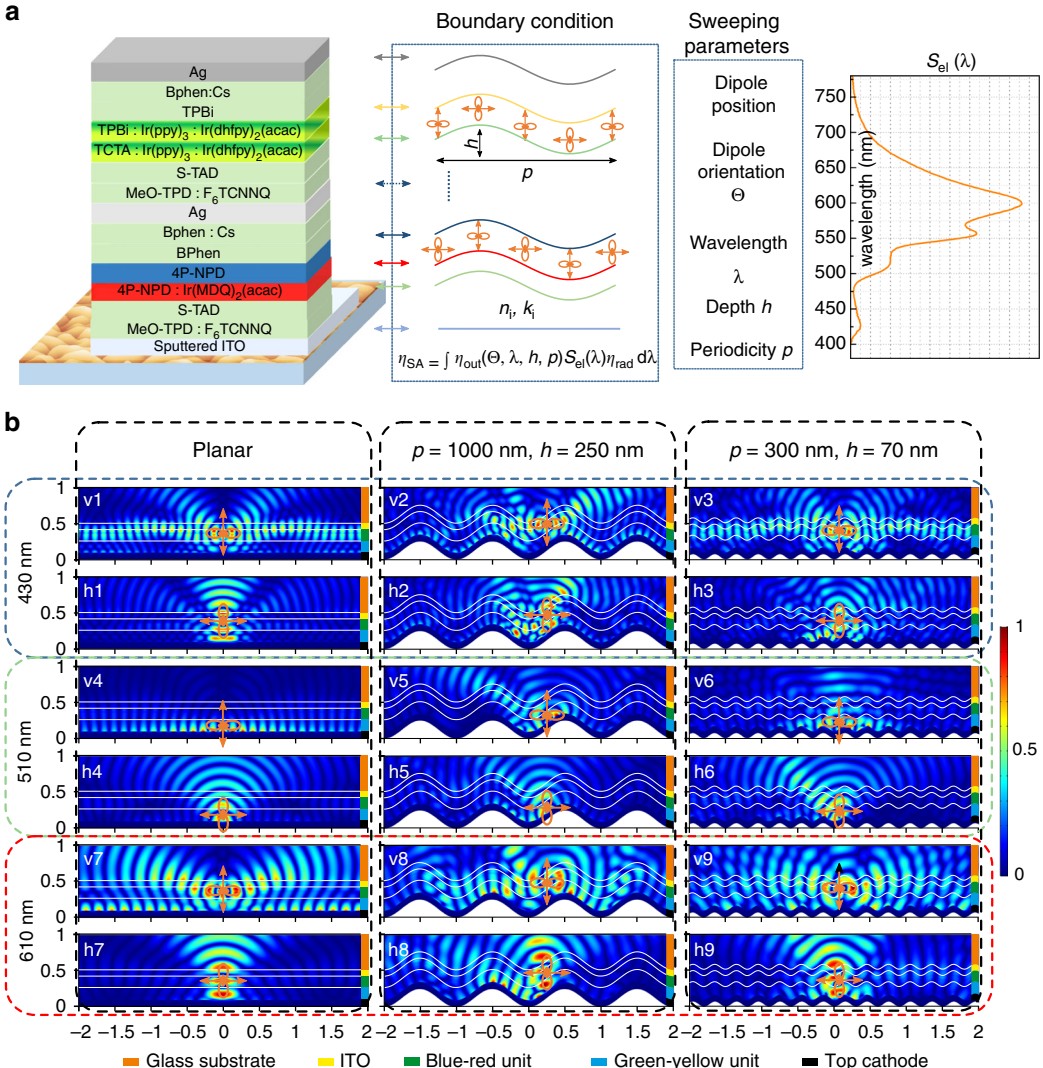

**Fig. 4** Schematic diagrams of the proposed optical modeling. **a** The device architecture and schematic illustration of the simulation model. The outcoupling efficiency $\eta_{out}$ is dependent on the dipole orientation $\Theta$ and the position on the sinusoidal interface, emitting wavelength $\lambda$, periodicity $p$, and depth $h$ of the sinusoidal nanostructures. **b** Normalized steady state electric field for vertical and horizontal dipoles. At wavelengths 430 nm for vertical dipoles (v1–v3) and horizontal dipoles (h1–h3), 510 nm for vertical dipoles (v4–v6), and horizontal dipoles (h4–h6), 610 nm for vertical dipoles (v7–v9) and horizontal dipoles (h7–h9). Three different configurations are presented – flat (first column), nanostructured devices with $P = 1000$ nm, $h = 250$ nm (second column) and $P = 300$ nm, $h = 70$ nm (third column). Dipoles are located in the middle - between minima and maxima of the sine texture inside corresponding emission layers. All dimensions are in μm

weighting the outcoupling efficiency with the normalized electroluminescent spectrum $S_{el}(\lambda)$ and multiplying with the effective radiative efficiency $\eta_{rad}$ of the planar white device, corresponding to experimental measurements of $\eta_{SA}$. The final $\eta_{SA}$ is calculated as an averaged result over five different dipole positions within one periodicity of sine texture for both horizontal and vertical dipoles[13,47] (See Methods and Fig. 4a). In principle, the final outcoupling performance of each random nanostructure is the sum of different periodicities and heights, corresponding to the periodicity and depth distribution for each of the nanostructure detected by the AFM measurements. As a first estimation, a sinusoidal texture with the dominant periodicity and dominant depth given by measured distributions can be representative for the final device performance based on quasi-periodic nanostructures.

To illustrate the light distribution inside the device, we show color maps of the normalized electric field for the planar device in Fig. 4b. We can notice that most photons emitted from the vertical dipoles are trapped in organic layers and at the surface of the metal cathodes. On the other hand, energy trapping for horizontal dipoles is less pronounced. Hence, the enhancement caused by the nanostructure is expected to be more significant for vertical dipoles compared to the horizontal ones. Moreover, the light distribution depends strongly on the position of the emitting dipoles in the stack, as we can see that main losses for the blue/red is due to waveguiding, since the emitting layers are close to the ITO and far from the top metal cathode. However, for the green/yellow emitting dipoles, which are much closer to the metal cathode, the main losses arise from the coupling of light to SPP modes. In Fig. 4b, we also show the normalized electric field of horizontal and vertical dipoles at three representative wavelengths (emission peaks), on two nanostructures with periodicity of 1000 nm and height of 250 nm, and with periodicity of 300 nm and height of 70 nm, respectively. We show here only dipoles positioned at the middle of the sine nanotexture, but it should be noted that the dipole position on the nanostructure strongly

influences on the outcoupling efficiency $\eta_{out}$. For example, the vertical dipole located at the bottom of the sine texture with a wavelength of 510 nm exhibits a value $\eta_{out}$ of 37.5% while at the middle of the sine texture it can reach up to 52.8%. Similarly, for the horizontal dipole, it reaches 55.5% and 71.7% at the bottom and middle of the sine texture, respectively. As there can be more than 15% absolute difference in $\eta_{out}$ between different dipole positions on the nanostructure, five simulated positions are taken into account, e.g., dipole in valley, on and between hills, allowing to simulate an uniform distribution of emitting molecules on the textured surface. More details on influence of dipole position and emitting wavelength are shown in Supplementary Fig. 5.

Numerically, the enhancement of $\eta_{SA}$ is dependent on the dipole orientation, the size of the nanostructure, the position of dipoles on the nanostructure and the radiated frequency, and emitting spectrum. In Fig. 5a, b, we show enhancement factors normalized to the peak intensity of the flat device, for textures with $p = 300$ and 1000 nm. For the white OLEDs based on nanostructures with periodicity $p = 300$ nm, the highest enhancement factor of ~1.35 can be obtained at a wavelength 600 nm, with a texture depth of about 70 nm. At 550 nm, the intensity of the planar device is reduced by a factor of ~0.75, while the intensity of the devices on a sinusoidal nanostructure can still reach up to 1.0. Thus, it is also possible to obtain a similar enhancement factor (1.0/0.75 = 1.33) at 550 nm. When increasing the periodicity to 1000 nm, shown in Fig. 5b, maximum enhancement factor of 1.40 at 600 nm can be realized with a sine texture depth of 220 nm. An even slightly higher enhancement factor can be achieved at 550 nm (1.20/0.75 = 1.6). The change of periodicity and depth at each wavelength can have influence on the enhancement factor. More periodicity/height variations are summarized in Supplementary Fig. 6. This is consistent with the general idea that the device efficiency depends on the geometry of the nanostructure and the emission wavelength. Moreover, a sinusoidal nanostructure with only one fixed periodicity can already induce a wavelength-dependent enhancement. This shows the advantage of using textures with a periodicity and depth distribution for white OLEDs, as these can contain a wider range of periods and heights and thus provide a more uniform enhancement over the entire emission wavelength.

In Fig. 5c it is demonstrated that the simulated enhancement factor is highly dependent on the aspect ratio AR of sinusoidal textures. Rigorous simulations show a distinct trend of improving device efficiency by increasing the AR up to ~0.25, where a maximum improvement of ~1.45 is predicted. Increasing the AR further decreases the efficiency. Moreover, higher efficiencies are predicted by using periodicities in the range between 500 nm and 1000 nm, while for a sine texture with a periodicity smaller than 300 nm (dimensions near subwavelength range) or larger than 1500 nm (structures are becoming flat in the dipole vicinity), improvements are less pronounced.

**White OLEDs on nanostructures**. RIE-induced nanostructures are applied for extracting trapped photons from bottom-emitting white OLEDs, as shown in Fig. 4a. Although these devices can in principle directly be fabricated on the PDMS surface, here we use a replica for different measurements and device fabrication (see Methods). The average transmittance of the sputtered ITO on the optical resin within the visible wavelength is about 76% (Supplementary Fig. 7). Here, we apply five different nanostructures named N1–N5 with varied periodicity distribution and depth distribution (Supplementary Fig. 10). Identical flat devices are fabricated on the same sputtered ITO for comparison.

As shown in Fig. 6a, under low driving voltage, the difference of voltage–current density curves mainly arises from the leakage current (Supplementary Fig. 8), resulting from the perturbation of

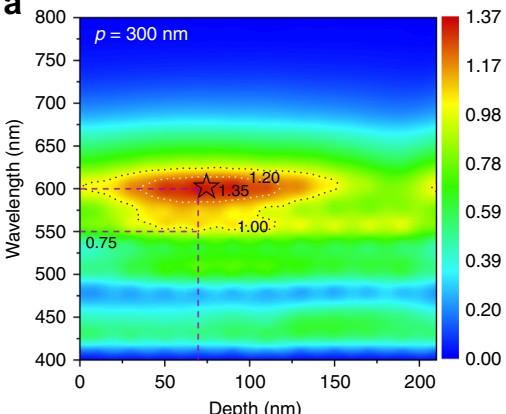

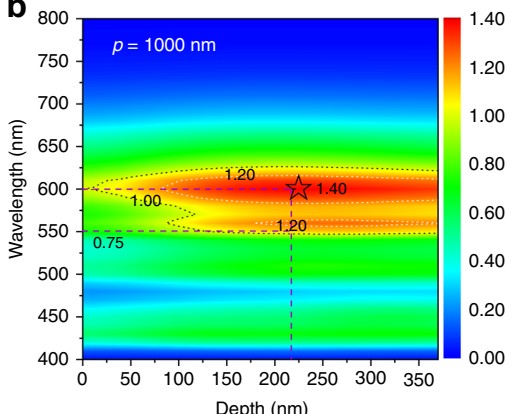

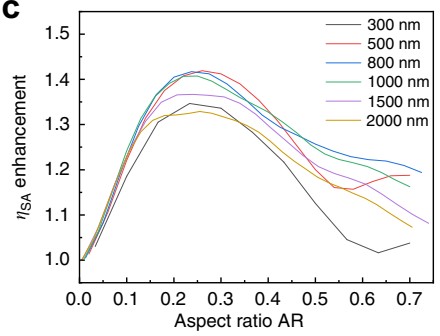

**Fig. 5** Numerical simulation of the enhancement for white OLEDs. The depth of 0 nm indicates flat devices. The spectra for flat devices are normalized as [0, 1]. The simulated intensity for devices on nanostructures at different wavelength is then normalized to the maximum intensity of the flat device. **a** White OLEDs on nanostructures with periodicity $p = 300$ nm. **b** $p = 1000$ nm with different depth are presented. **c** Numerical simulation results of $\eta_{SA}$ enhancement factor dependent on the aspect ratio of sinusoidal nanostructures with different periodicities

the nanostructures beneath the ITO and the intrinsically higher roughness of ITO films without annealing process[48,49]. In present investigation, the voltage–current density characteristics for all devices are identical for voltages larger than 6 V, since the influence from the leakage current is not significant in this range.

Previously reported nanostructured OLEDs show higher current densities under the same driving voltage compared to the planar device, because of the partially reduced distance between the peak and valley of the nanostructured bottom and top electrode[19,50]. However, this is not observed in our

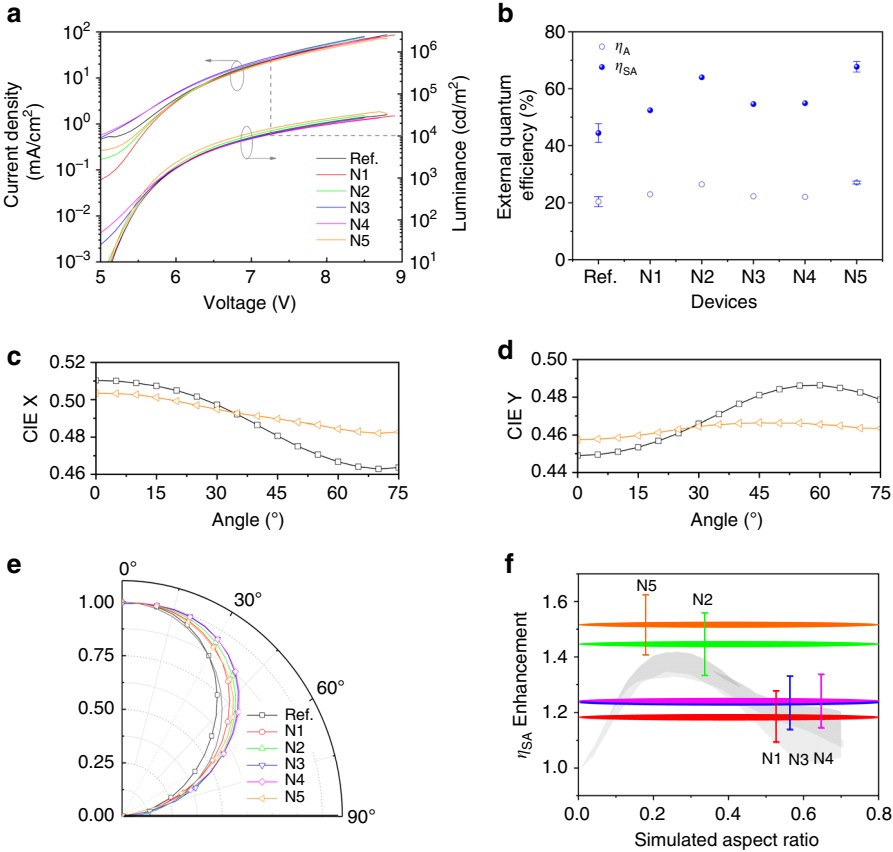

**Fig. 6** The performance of white OLEDs based on RIE-induced nanostructures. **a** Current density–voltage–luminance characteristics. **b** The $\eta_A$ and $\eta_{SA}$ at the luminance of 10,000 cd m$^{-2}$. **c** The angular dependence of CIE x. **d** The angular dependence of CIE y. **e** The angular radiant intensity. The gray line is the Lambertian distribution. **f** Dependence of the $\eta_{SA}$ enhancement factor on the aspect ratio of sinusoidal nanostructures. The gray band represents the range of simulated enhancement factors (cf. Fig. 5c). The error of $\eta_{SA}$ enhancement factor is obtained by averaging different devices. The horizontal lines reflect the problem to align the complex nanostructures with absolute certainty to the simulated ARs. The presented location refers to the case, where the experimental AR is calculated with the dominant depth $h$ as the AR-defining depth (see Supplementary Note 4 for details)

investigations, which might result from the utilization of p- and n-doped transport layers in this study as they possess much higher charge carrier mobility compared to the intrinsic transport materials[9]. The thickness reduction of the doped transport layers has little influence on the carrier transport and recombination processes[9,51,52]. Therefore, the efficiency enhancement at high luminance (see Fig. 6b, f) arises from the optical effect of the nanostructure and not from the thickness reduction of functional layers or change of the electrical efficiency.

To verify the influence of different nanostructures on the device performance, quantum efficiency $\eta_A$ and $\eta_{SA}$ are measured by a calibrated integrating sphere for all devices, presented in Fig. 6b and Supplementary Fig. 8. The planar device shows a maximum $\eta_A$ of 22.2 ± 3.1%. For devices with nanostructures, a maximum $\eta_A$ of 29.1 ± 1.1% can be obtained. The shape of the EQE versus luminance characteristics is influenced by the leakage current for these samples, which mainly influences the maximum value of $\eta_A$, rendering a comparison at low to medium luminance levels improper (Supplementary Figs. 8 and 12). At a luminance of 10,000 cd m$^{-2}$, where the influence of leakage current is negligible, the $\eta_A$ of the planar device slightly drops to 20.4 ± 1.8%, while it remains 27.3 ± 0.3% for the textured device N5. For the other textured devices, the $\eta_A$ can stay as high as 23–27% at 10,000 cd m$^{-2}$, as summarized in Fig. 6b.

The maximum $\eta_{SA}$ of 48.3 ± 5.8% can be obtained for the planar device and it rolls-off to 44.4 ± 3.3% at 10,000 cd m$^{-2}$. For textured samples based on nanostructures, a maximum $\eta_{SA}$ of

76.3% and luminous efficacy of 95.7 lm W$^{-1}$ is achieved and it rolls-off to 69.0% and 73.9 lm W$^{-1}$ at 10,000 cd m$^{-2}$. It demonstrates an enhancement factor of 1.53 ± 0.12 at 10,000 cd m$^{-2}$. Considering the influence of the leakage current, together with the mode distribution for the planar white OLEDs (Supplementary Note 2, Supplementary Fig. 11, and Supplementary Tables 7 and 8), we estimate the efficiency of light outcoupling structures (ELOS, Supplementary Note 3) for these nanostructures to be as high as 36.6% based on highly optimized white OLEDs[53].

It is interesting to note that the ratio of $\eta_{SA}$ to $\eta_A$ is higher for textured devices compared to planar devices, as summarized in Supplementary Table 5. For example, the ratio of $\eta_{SA}/\eta_A$ at 10,000 cd m$^{-2}$ for the planar device is 2.18 compared to 2.47 for device N5 with the nanostructure. This result indicates that the nanostructures couple more photons to the substrate which are then extracted by the attached hemisphere. The wavy profile of nanostructures can indeed guide the photons trapped as waveguide modes by reducing the incident angle to the substrate. However, because of the intrinsically flat geometry of these nanostructures (low AR), the incident angle is still high when transmitted to the interface of the glass substrate and air zone, leading to a situation where some of the extracted photons from waveguide modes or SPP modes are still in the substrate. Those photons can be rather easily extracted from the substrate with common external outcoupling structures. A similar phenomenon has been reported when using an ITO nanomesh for improved outcoupling from bottom green OLEDs[52].

As shown in Supplementary Fig. 9, there is no significant spectral change at different angles for devices with or without nanostructures. The shift of Commission Internationale de L'Eclairage (CIE) coordinates for different angles is depicted in Fig. 6c, d. A more pronounced CIE shift is noted for the planar device at different observation angles, while there is no significant CIE shift for the textured device, demonstrating that the incorporation of nanostructures into the white OLEDs improves the color stability.

Figure 6e shows the angular dependent radiant intensity for these devices with or without nanostructures. The emission profile is tuned from slightly less-Lambertian for the planar device to super-Lambertian emission for nanostructured devices. The angular and wavelength independent emission behavior demonstrates that the presence of quasi-periodic nanostructures can reduce the microcavity effect and increase the homogeneity of the energy distribution in the forward radiated hemisphere. RIE-induced nanostructures with a dominant periodicity reported in the present work, bring none of the drawbacks such as wavelength or angular dependent emission compared to 1D or 2D grating structures, which is important for lighting applications[18].

**Deviation between experimental and simulated enhancement**. We have seen an increased $\eta_{SA}$ for nanostructured white OLEDs compared to flat devices. For nanostructure N1–N4, the periodicity distribution ranges from <100 nm to more than 1000 nm, while N5 has a broader periodicity distribution to more than 3000 nm, peaking at ~1000 nm, as shown in Supplementary Fig. 10. There are several possibilities to define the experimental AR for these nanostructures, which are dependent on the different definition of the actual depth. A detailed analysis is given in Supplementary Note 4, to compare three cases: $2R_a$, full width at half maximum (FWHM) and dominant depth $h$ among the depth distribution. For such complicated nanostructure systems containing chaotic and ordered features, these possibilities allow to get a deeper understanding of the optical influence of these nanostructures.

We note that technically, when treating the dominant depth $h$ as the depth to calculate the AR in this study, a good match between experimental and simulation results of the enhancement factor can be obtained, even though the AR may vary in a broad range because of the broad depth distribution, as shown in Fig. 6f and Supplementary Note 4. The possible physical reasons are discussed in Supplementary Note 4. In this case (depth = dominant depth $h$), an AR ~0.2 shows the best device performance. When the AR is 0.60 (N4), the enhancement factor is 1.24 ± 0.10, growing towards an enhancement factor of 1.45 ± 0.12 obtained for a reduced AR of 0.41 (N2). The enhancement factor can be further increased to 1.53 ± 0.12 when AR drops to 0.19 (N5). These results indicate that the final enhancement is a synergistic effect from the periodicity and depth distribution of the nanostructures. As shown in Supplementary Table 6, both the absolute EQE and the enhancement factor for devices with the RIE-induced nanostructures reported here are among the top values compared to the reported results in the literatures.

The enhancement factor obtained from experimental results is slightly higher than in the numerical simulations, such differences can be assigned to the simplification of the simulation model to only two spatial dimensions because of a limited computing capacity, while texture and dipole are both 3D objects. On the other hand, in this 2D scenario, the directional orientation of such sinusoidal functions is not taken into account. Further improvement of the optical model to treat the nanostructures as 3D objects with the correct consideration of the in-plane periodicity distribution would be needed, which is beyond the scope of this work. Nonetheless, the fact that simulation and experimental trends of enhancement factor dependent in a

similar fashion on the aspect ratio demonstrates the rationality of simplifying quasi-periodic nanostructures to 2D sinusoidal textures. We anticipate that it is also possible to use this model to understand the optical effect of nanostructures on the device efficiency of perovskite light-emitting diodes[54].

## Discussion

We have demonstrated a method for extracting trapped photons from white OLEDs, by implementing quasi-periodic nanostructures induced by reactive ions etching on the PDMS surface. The topography of these nanostructures can be controlled by tuning the pretreatment conditions of the PDMS and the RIE treatment recipes. The mechanism for the nanostructure generation and control is explained by mechanical deformation within a bilayer system on a planar surface, initialized by compressive stress release because of external stimuli from chemical reactions, physical bombardments and the modulus mismatch between the RIE-induced silica-like stiff top layer and elastic bottom PDMS. The utilization of RIE-induced nanostructures in white OLEDs has shown the capability to efficiently extract waveguide modes and SPP modes leading to a higher efficiency, together with improved color stability and more homogeneous radiance distribution. An optical model considering the dipole position and dipole orientation is proposed to simulate the device performance by dividing the nanostructures into sinusoidal textures with a dominant periodicity and height. Optical simulations indicate that the highest enhancement can be expected for an aspect ratio of AR ≈ 0.25. Because nanostructures can be directly generated on the PDMS surface, they are compatible with emerging flexible devices. The controllable, facile and scalable method to fabricate these quasi-periodic nanostructures presents a powerful tool-set for generation and manipulation of complicated nanostructures, which also holds promising application potential in optical, biological, and mechanical fields.

## Methods

**Materials**. To eliminate the experimental error of mixing the base and curing agent, PDMS (SYLGARD® 184) with a fixed base and curing agent ratio of 10:1 is purchased from Sigma-Aldrich. For experiments to investigate the ratio influence and the pattern copy, PDMS (SYLGARD® 184) is purchased from Dow Corning, where the weight ratio of the base to curing agent can be varied. NOA 63 resist is purchased from Norland Products Inc. Perfluorodecyltrichlorosilane (FDTS) is bought from Alfa Aesar. Materials for OLED devices are purchased from Luminescence Technology Corp. and used after sublimation.

**PDMS preparation and RIE treatment**. The base and curing agent is mixed mechanically and then degassed in vacuum for 10 min. The mixture is then spin-coated on pre-cleaned glass substrates at 1000 rpm for 1 min. PDMS coated substrates are cured in an oven at varied temperatures for different heating time, explained in the following. After pretreatment, substrates are transferred to the RIE instrument (Oxford Plasmalab 80 Plus). After RIE treatment, samples are taken out to ambient environment with a humidity of 55% at room temperature. Samples for RIE power investigation shown in Figs. 2a, b, the weight ratio of the base to curing agent for PDMS is 10:1, pretreated at 80 °C for 80 min; The RIE recipe: 50 sccm $O_2$, 60 s. For pretreatment time investigation shown in Figs. 2c, d, the weight ratio of the base to curing agent is 10:1, prepared at 80 °C for 40 min, 60 min, 80 min, 120 min, and 160 min, respectively; The RIE recipe: 50 W, 50 sccm $O_2$, 60 s. For weight ratio investigation shown in Figs. 2e, f, the weight ratio of the base to curing agent is varied, pretreated at 80 °C for 80 min. RIE recipe: 50 W, 50 sccm $O_2$, 60 s.

**XPS measurement**. The chemical bonding states and atomic concentration are detected by using a XPS instrument (PHI 5600-CI, Physical Electronics, USA) with non-monochromatic Mg-Kα (1253.6 eV, 400 W), at the incident angle of 54°. The atomic concentration is calculated with standard single element sensitivity factors.

**DMA measurement**. DMA test is done by ARES2 (TA Instruments, USA). The size of PDMS samples are cured in petri dish and cut into bar shape. The size for each sample is calibrated by a micrometer, with a slight size variation from 4 cm × 1 cm × 1 cm. The measurement is done in a single-frequency scanning mode at 1 Hz, with a heating rate of 10 °C min$^{-1}$ over the temperature range from 60 to 100 °C.

**Pattern transfer**. As a demo, we here copy the patterns generated on PDMS surface as replicas for structure characterization and device investigation. It is possible to directly fabricate devices on top of the corrugated PDMS surface. The RIE-treated PDMS samples are vapor modified by FDTS for 24 h in a closed container in a glovebox. The PDMS mixture is used as the stamp material to copy patterns from RIE-treated PDMS samples. After mixing and degassing, the PDMS mixture is carefully poured onto FDTS treated samples and then annealed in an oven at 80 °C for 1 hour. The stamped PDMS can be easily peeled off and used as nanoimprinting stamps for the following device fabrication. Diluted NOA 63 by mixing with acetone at a weight ratio of 1:1 is spin-coated on cleaned glass substrates (size 25 mm × 25 mm) with a speed of 8000 rpm. The PDMS stamp is pressed into NOA 63 film by a home-made nano-imprinter and cured under UV radiation for 10 min.

**Topography measurement**. The pattern based on NOA 63 resist is measured by the atomic force microscope (AFM, AIST-NT Combiscope 1000, AIST-NT, Inc.), scanning electron microscopy (SEM, DSM 982, Carl Zeiss). To determine the periodicity distribution, a high scanning resolution of 1024 by 1024 in an area of 10 μm by 10 μm is selected. Further measurements at two different sites with a lower scanning resolution of 256 by 256 in an area of 10 μm by 10 μm are done to get the depth or height information. The periodicity distribution shown in this work is coming from the high-resolution scan and the depth is calculated from all the three measurements by averaging the results.

**ITO deposition**. Substrates with nanoimprinted NOA 63 is heated at 70 °C under vacuum for 5 h before ITO sputtering. The ITO anode is patterned as four-finger structures with a laser-cut metal mask. The ITO films are grown by sheet-to-sheet processing in the pilot scale in-line sputter coater. A conventional planar single magnetron system with oxide targets which is driven in direct current (DC) sputtering mode is used. The cathode length is 750 mm. The sputtering is done with a power of 3 kW and an additional oxygen gas flow of 6 sccm under a process pressure of 0.3 Pa. After sputtering, the ITO samples are annealed at 70 °C for 1 hour. The layer thickness of the ITO film is ~90 nm, with a sheet resistance of 67 Ω and a transmittance in the visible spectral range of 76%.

**OLED fabrication**. After the ITO sputtering, all glass substrates are used directly without any further cleaning process. After nitrogen blowing, substrates are heated under vacuum at 70 °C for 1 hour to get rid of moisture. All devices are made in a single-chamber Lesker tool (Kurt J. Lesker Co.) under vacuum of $10^{-7}$ to $10^{-8}$ mbar by thermal evaporation. Deposition rates are calibrated and monitored by quartz crystals. The white device consists of two units. The structures of the blue-red unit is: N,N,N′,N′-tetrakis(4-methoxyphenyl)-benzidine (MeO-TPD): 4 mol% 2,2′-(perfluoro-naphtha-lene-2,6-diylidene) dimalononitrile (F₆-TCNNQ) (35 nm)/2,2′,7,7′-tetrakis-(N,N′-diphenylamino)-9,9′-spirobifluorene (Spiro-TAD) (10 nm)/ N,N′-di-1-naphthalenyl-N,N′-diphenyl-[1,1′:4′,1″:4″,1‴-Quaterphenyl]-4,4‴-diamine (4P-NPD): 5 wt% Iridium(III)bis(2-methyldibenzo-[f,h]chinoxalin)(acetylacetonat) [Ir(MDQ)₂(acac)] (5 nm)/ 4P-NPD (4 nm)/ 4,7-diphenyl-1,10- phenanthroline (BPhen) (10 nm), the carrier generation layers are consisted of BPhen doped with cesium (90 nm)/Ag (0.5 nm)/MeO-TPD: 4 mol% F₆-TCNNQ (75 nm). The green-yellow unit is: Spiro-TAD (10 nm)/4,4′,4″-tris(N-carbazolyl)-triphenylamine (TCTA): fac-tris(2-phenylpyr-idine) iridium(III) [Ir(ppy)₃]: bis(2-(9,9-dihexylfluorenyl)-1-pyridine) (acetylacetonate) iridium(III) [Ir(dhfpy)₂ (acac)] (91:8:1 wt%) (5 nm)/ 2,2′,2″-(1,3,5-benzenetriyl)-tris[1-phenyl-1H-benzimidazole](TPBi): Ir(ppy)₃: Ir(dhfpy)₂(acac) (91:8:1 wt%) (5 nm)/TPBi (10 nm)/Bphen:Cs (60 nm)/Al (100 nm). After the top electrode deposition, devices are encapsulated in a glovebox under nitrogen atmosphere with UV curable glue and glass lids.

**Device evaluation**. The current density–voltage–luminance measurements are done with a KEITHLEY SMU2400 source-measure unit and a silicon photodiode at the same time. Electroluminescence spectra are taken by a calibrated spectrometer (CAS 140 CT, Instrument Systems). The external quantum efficiency and luminous efficacy are measured with a calibrated integrating sphere (LABSPHERE), with a SMU2400 and a calibrated spectrometer (CAS 140 CT). The angular dependent emission behavior is recorded by a custom-make goniometer with a calibrated spectrometer and a rotatory stage in a step resolution of 1°. The pixel size of OLED devices is calibrated with a standard OLED because of the slight variation of mask opening for the ITO sputtering, with the pixel size ranging from 6.7 to 8.2 mm².

**Device modeling**. To numerically simulate the loss channels, the planar tandem device is divided into two units, and each unit is simulated separately, while still having the other unit as passive layers present. The quantum efficiency for air modes, substrate modes, waveguide modes, and evanescent modes are summed up respectively, to get the fraction of photons in each mode for the tandem device, of which the ideal EQE would be 200%. As input data, layer thicknesses with corresponding complex refractive indices are used. Other data, like the anisotropy factor, the radiative efficiency and the electrical efficiency are taken from the literature[42,45]. More details can be found in Supplementary Note 2.

Optical simulations of corrugated devices are performed using a commercially available simulation tool Comsol Multiphysics based on the finite element method (FEM)[55]. Here, we treat emitting dipoles and nanostructures with the 2D model to reduce the computing load. The simulated device architecture is very similar to experimental devices with the same total thickness, in which thin layers sandwiching emission dipoles are treated as a single emission layer, to avoid the need for very small mesh elements in very thin layers ($d \leq 10$ nm). The optical indices ($n$, $k$) for each layer is set according to the experimental measured results. We use a sine structure with various periodicities and heights to simulate light outcoupling to the glass substrate, where the glass substrate is treated as a half-infinite medium. Organic emitting molecules are much smaller than light wavelength, thus emission sources in simulations can be treated as differently oriented point dipoles positioned at the corresponding emission interface. The simulated area in the model is set to a lateral size of 20 μm around the dipole, and the entire structure is surrounded by a perfectly matching layer (PML) as the absorbing boundary condition, to suppress any reflections at the boundaries. Collection of the emitted light is at least one wavelength away from the thin-film structure (flat and textured) to avoid coupling of evanescent waves to the PML.

The simulation is done in 10 nm wavelength steps from 400 to 800 nm, for sine textures with periodicities ranging from 300 nm to 2000 nm and different heights. For comparison, a flat structure is simulated, and internal efficiencies are set to match experimental results. Parameters in simulations are held constant during all simulations for nanostructured devices, with only varied values of $p$ and $h$ of the sinusoidal texture.

## Data availability
The data that support the findings of this study are available from the corresponding author upon reasonable request.

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

## Acknowledgements

Y.L. acknowledge the financial support of China Scholarship Council (No. 201506160049). L.J. acknowledge the financial supported from NSFC (No. 21506258), Natural Science Foundation of Hunan Province (No. 2016JJ3134), and China Scholarship Council (No. 201706375057). This work has received funding from the European Research Council (ERC) under the European Union's Horizon 2020 research and innovation program (grant agreement No 679213, project acronym BILUM), from the European Union's Horizon 2020 research and innovation program (grant agreement no. 646259; project acronym MOSTOPHOS and grant agreement no. 641725; project acronym PHEBE). The support from A. Hiess, B. Chang, and Jinxiu Yu is acknowledged here. We also thank Prof. Karl Leo for beneficial discussion.

## Author contributions

Y.L. conceived the idea, designed, and conducted experiments, and wrote the manuscript. Y.L., M.K., P.W., and C.H. contributed to the optical modeling. Y.L., L.J., and Z.M. made contributions to the mechanism of nanostructure generation. Y.L. and S.O measured the XPS and made the data analysis. Y.L., J.W., and M.J. made the ITO sputtering. Y.L. fabricated and tested the OLED devices. Y.L., P.W., S.L., R.S., and S.R. analyzed the device results. S.L. and S.R. organized the entire project. All the authors contributed to the manuscript.

## Additional information

**Competing interests:** The authors declare no competing interests.

