## [Peer Review File · Nature Communications]

Reviewers' comments:

Reviewer #1 (Remarks to the Author):

In this work, the authors report some interesting results on highly efficient OLED leveraging a new out-coupling scheme. It is well-known that today OLED can have an internal quantum efficiency approaching 100%. However, the external quantum efficiency is usually 20 to 40% (a recent paper demonstrated an external efficiency of >50%), due to the photon trapping effect. Some of the photons are trapped by total internal reflection, and some are trapped by the waveguiding effect.

In this work, the authors demonstrate an external efficiency above ~70% by using a pre-patterned substrate with nanostructures. Overall the results are quite sound. However, I do have a few comments regarding the performance of the OLEDs.

1. The reported efficiency is plotted in Fig. 6b. It seems that the external efficiency varies quite significantly. The lowest is about 45% and the highest is about 70%. What is the cause of this large variation? Is the reported ultrahigh efficiency reproducible and uniform across a large area?
2. The nanostructures are patterned using RIE. Those nanostructures are seemed to be very random. I am wondering how reproducible the nanostructures are. If devices are produced by multiple runs, do they exhibit similar performance?
3. Finally, I suggest the authors to use a table to summarize the external quantum efficiency reported by a few top groups and companies and this work. Then the advantage of this reported outcoupling scheme can be clearly seen.

Reviewer #2 (Remarks to the Author):

This manuscript reports on the fabrication of random nanostructures and their use as an internal light-extraction layer in OLEDs. The authors present a recipe how to modify the average "periodicity" and depth of the nanostructures over a relatively large range. They furthermore present a simulation-based analysis of the light extraction capability of such a layer. And, they apply it to tandem white OLEDs to demonstrate the improved light outcoupling experimentally. Technically, this is a sound paper and there is not so much to criticize about the contents. Just a few issues should be addressed:

- The formation of random nanostructures and their use in OLEDs was demonstrated many years before (see e.g. Ref. 19 of the manuscript). And there is even older work demonstrating that thin films can spontaneously form such patterns upon inducing morphological instability by certain means. Thus, the main claim here is to have a new, technically easier recipe to achieve this.
- The reported external quantum efficiency of 74% is achieved using a bulky macro-extractor, which is not consistent with the thin appearance of OLEDs. And one has to keep in mind that the device is a tandem OLED, where the EQE is effectively doubled. This means that compared to reported values on single-emission unit OLEDs with light extraction schemes preserving the thin-film device nature, the EQE of this work is nothing exceptional.
- The way the manuscript is written is not always appealing. I suggest shifting some of the more technical stuff to the SI.

Reviewer #3 (Remarks to the Author):

How to enhance the outcoupling efficiency in white OLEDs is an important topic in OLED lighting field. As we know, OLEDs suffer from notorious light trapping, resulting in that the efficiency of the resulting OLEDs can be further enhanced, hindering its application. In this paper, the authors developed a facile, scalable and lithography-free method to generate controllable nanostructures with directional randomness and dimensional order by reactive ions etching (RIE) to enhance the outcoupling efficiency of the fabricated white OLEDs. As we see, about 1.3 times enhancement is

obtained. They gave the detailed results about experimental conditions that are used to obtain the nanostructure, and optical simulation. The method to obtain the nanostructures is significant and useful and the results are also encouraging. However, the obtained nanostructures still exist some problems that need to be considered by the authors.

1. As given, the nanostructures is directional randomness. This means that the repeatability can not be guaranteed. If so, the simulation are also unreliable.
2. As we see, the fabricated OLEDs exhibit large leakage current due to the large surface roughness, which will greatly influence device efficiency and stability, causing poor applicability.
3. The 1.3 times enhancement in efficiency is not high as internal outcoupling structure.
4. White OLEDs as lighting sources in practical applications are required to have big emission area. Can the nanostructures given in this paper keep the uniformity and repeatability in big area?

Response to Reviewers' Comments

We sincerely thank the three reviewers for their thoughtful comments and constructive suggestions for improving the manuscript. Responses to each of the comments are summarized below. The original comments are *italic*, our responses in 'normal' typeset. Sections describing changes during the revision are highlighted here. The corresponding changes in the manuscript and supplementary information are marked as blue in the revised version. As new figures and tables are added to the supplementary information during this revision, the number of figures and tables are re-organized in the revised version.

At the end of this response letter, we have added a paragraph in purple, discussing a formal change that is not in connection with the reviewers' comments.

Reviewer #1:

In this work, the authors report some interesting results on highly efficient OLED leveraging a new out-coupling scheme. It is well-known that today OLED can have an internal quantum efficiency approaching 100%. However, the external quantum efficiency is usually 20 to 40% (a recent paper demonstrated an external efficiency of >50%), due to the photon trapping effect. Some of the photons are trapped by total internal reflection, and some are trapped by the waveguiding effect.

In this work, the authors demonstrate an external efficiency above ~70% by using a pre-patterned substrate with nanostructures. Overall the results are quite sound. However, I do have a few comments regarding the performance of the OLEDs.

Reviewer #1's Comment#1:

1. The reported efficiency is plotted in Fig. 6b. It seems that the external efficiency varies quite significantly. The lowest is about 45% and the highest is about 70%. What is the cause of this large variation? Is the reported ultrahigh efficiency reproducible and uniform across a large area?

Authors' response to Reviewer #1's Comment #1:

Thank you for this comment. Please note that the devices giving ~45% external quantum efficiency (EQE) are references without nanostructure (the value is so high, because they are tandem structures with up to potentially 200% internal quantum efficiency). The devices with nanostructures N1-N5 yield higher EQEs compared to the reference, but the absolute EQE is dependent on the specific nanostructure, as shown in Figure 6b in the main text. The highest EQE ~70% can be obtained for devices with the nanostructures N2 and N5. The efficiencies are obtained for tandem white OLEDs with an attached half-sphere lens to extract substrate modes, so actually it is quite reasonable, which is also pointed out by the 2nd reviewer (see below).

The efficiency is reproducible since we observed several samples with the highest EQE ~ 70%. The summary about the efficiency of these samples are presented in the supplementary information Figure S8 in the revised version.

With respect to the question of uniformity, we added supportive data to demonstrate the very good uniformity of the nanostructures generated on the PDMS surface. The uniformity investigation of the nanostructures is done by AFM measurements on different positions (at least four positions) randomly chosen for each sample, followed with a statistical analysis to check the variation of the obtained values measured at each position. As shown in Figure S1, the periodicity distribution of a specific structure (N1) is almost the same for all these measurements. The dominant periodicity for N1 is 245.5 nm. There is negligible deviation of the average depth at different positions as shown in Figure S1. This originates from the intrinsic difference at different positions and the experimental deviation for each AFM measurement.

Figure S1. Uniformity investigation on sample N1 as a representative nanostructure generated by RIE treatment on PDMS. The structure is investigated with AFM at different local positions (P1 – P4). (a) Periodicity distribution. (b) Average depth. The AFM measurement is done with a resolution of 1024×1024.

The parameters describing the nanostructure, i.e. the full width at half maximum (FWHM) and the dominant depth for the depth distribution of all investigated nanostructures at different positions, are summarized in Table S1 and Table S2 in the response letter. According to the statistical analysis, the deviation of the proposed aspect ratio (AR, $AR = \text{depth} / \text{periodicity}$) for the nanostructures at different positions is very small, indicating that the nanostructures are uniform at different positions across the entire surface.

Table S1. The deviation of the aspect ratio based on the FWHM of the depth distribution determined from multiple AFM measurements at different sample positions. The dominant periodicity is obtained from AFM measurements with a resolution of 1024×1024. The proposed aspect ratio is calculated as FWHM / dominant periodicity.

Sample	FWHM of depth distribution (nm)						Dominant periodicity (nm)	Mean of AR	Standard deviation
	256×256			1024×1024					
N1	73.6	73.7	90.3	81	77.8	84.1	245.5	0.326	0.026
N2	90.9	91.6	-	99.9	96.1	95.6	449.9	0.211	0.008
N3	83.7	89	-	92	82.9	-	245.5	0.354	0.018
N4	128.1	141.5	-	159.9	132	-	337.5	0.416	0.042
N5	149	149	-	155.8	150.8	-	1043	0.145	0.003

Table S2. The deviation of the aspect ratio based on the dominant depth of the depth distribution determined from multiple AFM measurements at different sample positions. The dominant periodicity is obtained from AFM measurements with a resolution of 1024×1024. The proposed aspect ratio is calculated as dominant depth/dominant periodicity.

Sample	Dominant depth of depth distribution (nm)						Dominant periodicity (nm)	Mean of AR	Standard deviation
	256×256		1024×1024						
N1	123.4	110	142.9	137.8	126.2	136.1	245.5	0.527	0.049
N2	148.4	155.5	-	161.7	134	158.3	449.9	0.337	0.024
N3	141	122.8	-	149.8	140.3	-	245.5	0.564	0.046
N4	184.9	209.4	-	263.4	216.5	-	337.5	0.648	0.097
N5	213.5	157.5	-	202	175	-	1043	0.179	0.024

We believe that substrates with larger size should also work, since the RIE treatment is now a mature technology used for many applications such as chip fabrication.¹ In our case, the RIE setup (Oxford Plasmalab 80 Plus) is suitable for substrates with maximum size up to 200 mm (data from the manufacturer).²

The uniformity investigation is now also added to the supplementary information as Figure S2, Table S1 and Table S2 and explaining paragraph in the revised version.

Reviewer #1's Comment #2:

The nanostructures are patterned using RIE. Those nanostructures are seemed to be very random. I am wondering how reproducible the nanostructures are. If devices are produced by multiple runs, do they exhibit similar performance?

Authors' response to Reviewer #1's Comment #2:

This is a very good comment. The reproducibility and uniformity have been paid special attention in the early stage of this study.

We have checked on the experimental repeatability of these nanostructures by using a tracking sample made with the same fabrication recipe for each run of RIE treatment. Further this data was collected to check the setup stability.

As presented in Figure 1 in the revised manuscript and Figure S1 in the supplementary information, the nanostructure is randomly orientated on the surface of PDMS after RIE treatment, but statistically, there is a distribution of the periodicity and depth for each nanostructure. The nanostructure is a system bridging chaos and order as a quasi-periodic pattern. According to AFM measurements, one can obtain the statistical properties including the periodicity, depth distribution and average depth. The experimental repeatability is monitored by measuring the quantitative parameters of periodicity and depth for the nanostructure generated in different batches with the same recipe.

Figure S2 Experimental variation of the nanostructures of samples prepared with the same recipe during multiple, consecutive runs (batches 1-5). (a) Periodicity distribution. (b) The average depth. The error bar is the standard deviation from at least three AFM measurements at different positions for each sample. The PDMS preparation recipe: PDMS from Sigma-Aldrich, 1000 rpm, 80°C, 80 min. The RIE recipe: 50 sccm O₂, 50 W, 60 s.

As shown in Figure S2 in this response letter, for nanostructures generated in multiple runs, the periodicity is peaking at ~350 nm. Only very small deviation of the average depth can be noted, as shown in Figure S2b. Therefore, the method we presented in this study to generate nanostructures is controllable with good repeatability.

Now we add this data set as Figure S3 and an explaining paragraph in the supplementary information.

Reviewer #1's Comment #3:

Finally, I suggest the authors to use a table to summarize the external quantum efficiency reported by a few top groups and companies and this work. Then the advantage of this reported outcoupling scheme can be clearly seen.

Authors' response to Reviewer #1's Comment #3:

This is a very good comment. Thank you very much. Having this included will put our study in better context.

Table S3. Comparison of internal outcoupling structures for white OLEDs reported in literature. Improvements are calculated based on published values of external quantum efficiency (EQE) of white OLEDs without any outcoupling strategy (w/o), with only internal outcoupling strategy (w/in), with only external outcoupling strategy (w/ex), and with both internal and external outcoupling strategy (w/w). The enhancement factor is calculated in three different cases I: EQE(w/w) / EQE (w/o), II: EQE (w/in) / EQE(w/o), III: EQE (w/w) / EQE (w/ex) in this summary, depending on the data availability.

Light extraction strategies		External quantum efficiency (%)				Enhancement factor	Ref.
		w/o	w/in	w/ex	w/w		
Internal	External						
Low index grid ^{a,*}	Micro lens array	14.7	19	25	34	2.31 ^I , 1.29 ^{II} , 1.42 ^{III}	[3]
Deterministic aperiodic nanostructures (DANs) ^{a,*}	-	26	56	-	-	2.15 ^{II}	[4]
Vacuum nanohole array (VaNHA) ^{b,*}	Half-sphere lens	19.3	43.9	36.9	75.9	3.93 ^I , 2.27 ^{II} , 2.06 ^{III}	[5]
High index substrate ^{a,*}	Index-matched half-sphere lens	13.1	14.4	24	34	2.60 ^I , 1.10 ^{II} , 1.42 ^{III}	[6]
Nano-particle based scattering layers (NPSLs) ^{b,†}	Half-sphere lens	22	33	-	46	2.09 ^I , 1.5 ^{II}	[7]
Subelectrode micro lens array (SEMLA) ^{a,*}	Micro lens array	16	20		27	1.69 ^I , 1.25 ^{II}	[8]
Multifunctional Nanofunnel Arrays (NFAs) ^{b,*}	NFAs	12.7	20	19.6	29.4	2.31 ^I , 1.57 ^{II}	[9]
Metal oxide nanostructures ^{c,†}	Half-sphere lens	14.3	20.3	26.6	35.5	2.48 ^I , 1.42 ^{II} , 1.33 ^{III}	[10]
RIE-induced nanostructures ^{c,†}	Half-sphere lens	20.4	27.3	44.4	69.0	3.38 ^I , 1.34 ^{II} , 1.55 ^{III}	This work

a, averaged peak value, if available. b, at 1,000 cd m⁻². c, average value at 10,000 cd m⁻².

*, single-unit white OLEDs. †, double-unit tandem white OLEDs

Now we add a table in the supplementary information as Table S6 (named as Table S3 in this response letter), to compare the external quantum efficiency (EQE) of white OLEDs and the enhancement factor from different studies. As shown in Table S3 in this response letter, both the absolute EQE and the enhancement factor for devices with the RIE-induced nanostructures in this study are among the top values compared to the reported results in the literatures, demonstrating the efficiency of these nanostructure to couple out trapped photons from white OLEDs.

Reviewer#2:

This manuscript reports on the fabrication of random nanostructures and their use as an internal light-extraction layer in OLEDs. The authors present a recipe how to modify the average “periodicity” and depth of the nanostructures over a relatively large range. They furthermore present a simulation-based analysis of the light extraction capability of such a layer. And, they apply it to tandem white OLEDs to demonstrate the improved light outcoupling experimentally.

Technically, this is a sound paper and there is not so much to criticize about the contents. Just a few issues should be addressed.

Reviewer #2’s Comment #1:

The formation of random nanostructures and their use in OLEDs was demonstrated many years before (see e.g. Ref. 19 of the manuscript). And there is even older work demonstrating that thin films can spontaneously form such patterns upon inducing morphological instability by certain means. Thus, the main claim here is to have a new, technically easier recipe to achieve this.

Authors’ response to Reviewer #2’s Comment #1:

It is true that quasi-periodic nanostructures have been used to demonstrate the success of outcoupling trapped photons in OLEDs. In Ref. 19 in the main text, the

nanostructures are generated by multiple times to obtain sufficient depth for realizing meaningful enhancement for bottom green OLEDs, since it is difficult by that methodology to tune the periodicity and depth independently. Here, in this work, we presented a new method to fabricate nanostructures with features ranging from nanometer to micrometer scales. The important finding is that process parameters can be used to deterministically fabricate tailored structure features – so the ‘control’ aspect of the process is the key here. An additional benefit is that this process is indeed, as Reviewer #2 noted, the speed and simplicity of the process. Specifically, it is possible to independently control the periodicity and depth, as illustrated in Figure 2 in the main text and Figure S4 in the supplementary information. The detailed analysis is summarized in the main text of the manuscript in the results part (section: nanostructure generation and characterization). In the end, we investigate the mechanism of nanostructure generation and control, which is presented in the main text (section: mechanism of the nanostructure control). To sum up, we presented a facile, scalable, lithography-free, reproducible and controllable method to fabricate nanostructures with sizes ranging from nanometers to micrometers.

Reviewer #2’s Comment #2:

The reported external quantum efficiency of 74% is achieved using a bulky macro-extractor, which is not consistent with the thin appearance of OLEDs. And one has to keep in mind that the device is a tandem OLED, where the EQE is effectively doubled. This means that compared to reported values on single-emission unit OLEDs with light extraction schemes preserving the thin-film device nature, the EQE of this work is nothing exceptional.

Authors’ response to Reviewer #2’s Comment #2:

The investigation in this study is based on a highly optimized tandem white OLED. In theory, the maximum internal quantum efficiency for the reference should be 200%. According to the simulation of the outcoupling efficiency for the planar tandem OLED

in the Supplementary note 1, the internal quantum efficiency is about 125%, which is consistent with our previous investigation on that device architecture. (Ref. 45 in the main text). The detailed analysis for the quantification of dissipation modes including air modes, substrate modes, waveguide modes and surface plasmon polariton (SPP) modes can be found in Supplementary note 1. Since the half-sphere lens can extract substrate modes normally trapped inside the substrate, the reference device without RIE generated nanostructure can already give the maximum external quantum efficiency (EQE) of $48.3 \pm 5.8\%$. The EQE of devices with nanostructures can reach 69.0 % at $10,000 \text{ cd m}^{-2}$. The detailed analysis is presented in the result part in the main text (section: white OLEDs on nanostructures).

With respect to the comment on the use of the half-sphere lens as macro-extractor, we used it only to get access to all power that is otherwise trapped in the substrate. This is important because the RIE nanostructures will have a different coupling from organic to substrate modes (different intensity and angular distribution) compared to the reference device.

Reviewer #2's Comment #3:

The way the manuscript is written is not always appealing. I suggest shifting some of the more technical stuff to the SI.

Authors' response to Reviewer #2's Comment #3:

Thank you very much for this feedback, which we believe it is mostly with respect to the detailed description of the RIE process and its variations on the produced structures. Here, we would like to not follow the suggestion of Reviewer #2, to shift some of this description to the SI, as we strongly believe that these details are vital in differentiating our work from earlier reports on nanostructure generation. Hence, we decided to keep our manuscript structure as it is.

Reviewer #3:

How to enhance the outcoupling efficiency in white OLEDs is an important topic in OLED lighting field. As we know, OLEDs suffer from notorious light trapping, resulting in that the efficiency of the resulting OLEDs can be further enhanced, hindering its application. In this paper, the authors developed a facile, scalable and lithography-free method to generate controllable nanostructures with directional randomness and dimensional order by reactive ions etching (RIE) to enhance the outcoupling efficiency of the fabricated white OLEDs. As we see, about 1.3 times enhancement is obtained. They gave the detailed results about experimental conditions that are used to obtain the nanostructure, and optical simulation. The method to obtain the nanostructures is significant and useful and the results are also encouraging. However, the obtained nanostructures still exist some problems that need to be considered by the authors.

Authors' response to Reviewer #3's general comment:

Thanks to Reviewer #3 for the positive feedback. Just to clarify the key findings in this manuscript, we presented a facile, scalable, lithography-free and controllable method to generate controllable quasi-periodic nanostructures, bridging a chaotic and ordered system. When embedded in a highly optimized tandem white OLEDs, it is possible to enhance the external outcoupling efficiency with substrate modes from $44.4 \pm 3.3\%$ to 69.0% at $10,000 \text{ cd m}^{-2}$, giving an enhancement factor of 1.53 ± 0.12 .

Reviewer #3's Comment #1:

As given, the nanostructures is directional randomness. This means that the repeatability can not be guaranteed. If so, the simulation are also unreliable.

Authors' response to Reviewer #3's Comment #1:

Thank you for pointing us – similar to Reviewer #1 – to the uniformity and repeatability of the reported nanostructure generation. The experimental repeatability is of vital

importance for every study and the reproducibility of nanostructures has been paid special attention in this work in the early stage.

As discussed in the main text in the results part (section: nanostructure generation and characterization), the geometry of the nanostructures is randomly orientated on the surface of PDMS after the RIE-treatment. Statistically, the dimensional parameters including the periodicity and depth is quasi-periodic with a distribution that can be measured by AFM. The experimental repeatability is monitored by measuring the periodicity distribution and average depth for the nanostructure generated in different batches with the same recipe as a tracking sample.

As shown in Figure S2 in this response letter (see above), in multiple batches from different time, the deviation of the dominant periodicity and the average depth of the nanostructure (tracking sample) generated with the same recipe is very small. For tracking samples generated in multiple runs, the periodicity is peaking at about 350 nm. Only very small deviation of average depth can be noted, as shown in Figure S2b. The method we demonstrated here is facile and controllable, with good experimental repeatability.

We made a detailed statistical analysis about the uniformity and repeatability for the nanostructure at different positions. **This has been added to the revised manuscript.** Please refer to the reply to Reviewer #1's Comment #2. To be short, the nanostructure is randomly oriented on the surface. However, regarding the quantitative parameters such as periodicity and depth, it is uniform statistically on the entire surface. Hence, when comparing to a simplified and fixed model description, we can guarantee that each individual structure can be seen fixed and reproducible for a given processing parameter set.

The optical model we proposed in this contribution is a 2-dimensional model. Thus, the randomly orientated nanostructures are simplified as a corrugated surface in a form of a sinusoidal function. By sweeping the periodicity and amplitude of the sinusoidal

function as well as the emitting wavelength, it is possible to calculate the outcoupling efficiency of devices with nanostructures compared to the reference. In this scenario, the directional orientation of the sinusoidal function is not taken into account at this stage. This deficiency has now been added into the main text and marked in blue in the revised version to remind the reader. This improvement should be considered in the further development of the 3-dimensional optical model.

Even though the model at this stage is only 2-dimensional, the rationality is maintained from several observations. Firstly, according to experimental results of the nanostructures, the surface profile is similar with the appearance of sinusoidal corrugations, as presented in Figure 1 in the main text. The simplification as sinusoidal functions in the optical model is consistent with the experimental results in terms of the geometrical characteristics of the RIE-nanostructures. Secondly, according to the simulation results based on the proposed model, a wavelength dependent enhancement can be noted for sinusoidal nanostructure with a single periodicity, as shown in Figure 5 and discussed in the results part (section: Optical modeling of white OLEDs on RIE-induced nanostructures). This is consistent with previous investigations (ref. 17 and 18 in the main text), that periodic nanostructures can induce wavelength dependent emission, reversely demonstrating that the model is reasonable. Finally, as shown in the Supplementary note 3, the simulated enhancement factor gives a similar trend as the experimental enhancement factor dependent on the aspect ratio (AR, $AR = \text{depth} / \text{periodicity}$).

It is a first step to numerically simulate the optical effect of the nanostructures with a distribution of periodicity and depth by the proposed model in this study. Further improvement of the optical model by treating the nanostructures as 3D species with the consideration of the directional arrangement of the corrugation would be meaningful. This statement has now been added into the main text in the revised version.

Reviewer #3's Comment #2:

As we see, the fabricated OLEDs exhibit large leakage current due to the large surface roughness, which will greatly influence device efficiency and stability, causing poor applicability.

Authors' response to Reviewer #3's Comment #2:

Thank you for this important comments.

The leakage current is about 1-2 orders higher compared to our previous reports with the same device structure on commercial ITO, as stated in the main text in results part (section: white OLEDs on nanostructures). It should be noted that in this study, the devices are fabricated with custom-made sputtered ITO without annealing on the photoresist named NOA 63. It is most likely both the perturbation of the nanostructures and the intrinsically higher roughness of ITO films without annealing process contribute to higher leakage current. The deficiency of the ITO anode should be one of the main sources leading to the high leakage current. As presented in the supplementary information Figure S8 in the revised version, the leakage current for the reference samples without the nanostructure is comparable to the structured device, demonstrating that the roughness of the flat ITO anode could already give rise to high leakage current.

The reason why there is a lack of thermal treatment of ITO in this study is resulting from the presence of photoresist NOA 63 under the anode. The photoresist is used to duplicate the nanostructure from the master mold, with a comparable refractive index to the glass substrate. The photoresist can only tolerate a thermal treatment with highest temperature about 60°C (data from the supplier).¹¹ As demonstrate by Ref. 48 in the main text, a higher annealing temperature can reduce the roughness of ITO and enhance the conductivity. **The discussion about the origin of high leakage current is now added in the revised version in the supplementary information.**

Reviewer #3's Comment #3:

The 1.3 times enhancement in efficiency is not high as internal outcoupling structure.

Authors' response to Reviewer #3's Comment #3:

The maximum EQE with substrate modes of $48.3 \pm 5.8\%$ can be obtained for the planar device and it rolls-off to $44.4 \pm 3.3\%$ at $10,000 \text{ cd m}^{-2}$. For textured samples based on nanostructures N5, a maximum EQE (with substrate modes) of 76.3% and luminous efficacy of 95.7 lm W^{-1} is achieved and it rolls-off to 69.0 % and 73.9 lm W^{-1} at $10,000 \text{ cd m}^{-2}$. It demonstrates an enhancement factor of 1.53 ± 0.12 at $10,000 \text{ cd m}^{-2}$. As summarized in Table S3 in this response letter, the enhancement factor in this study is among the highest for white OLEDs with internal outcoupling structures. The comparison of current outcoupling strategies for white OLEDs is now added into the supplementary information as Table S6.

It should be noted that the enhancement factor is dependent on the nanostructures, mainly the aspect ratio (AR) and the dominant periodicity. When the AR is 0.60 (N4), the enhancement factor is 1.24 ± 0.10 , growing towards an enhancement factor of 1.45 ± 0.12 obtained for a reduced AR of 0.41 (N2). The enhancement factor can be further increased to 1.53 ± 0.12 when AR drops to 0.19 (N5).

Reviewer #3's Comment #4:

White OLEDs as lighting sources in practical applications are required to have big emission area. Can the nanostructures given in this paper keep the uniformity and repeatability in big area?

Authors' response to Reviewer #3's Comment #4:

Thanks for this comment. This comment are very similar to the Comment #2 and Comment #3 from Reviewer #1. Please also refer to the reply of Reviewer #1's Comment #2 and Comment #3.

The nanostructures generated on the surface of PDMS is uniform in this study. This is confirmed by measuring randomly chosen positions on one sample and checking the deviation. As shown in Figure S1 in this response letter, the periodicity distribution of a specific structure (N1) is almost the same for all these measurements at different positions. The dominant periodicity for N1 is 245.5 nm. There is only a slight deviation of the average depth at different positions, shown in Figure S1b. This is originated from the intrinsic difference at different positions and the experimental deviation for each AFM measurement.

For all nanostructures investigated in devices, the statistical values including the full width at half maximum (FWHM) and the dominant depth among the depth distribution at different positions, are summarized in Table S1 and Table S2 in this letter. According to the statistical analysis, the deviation of the proposed aspect ratio (AR) for these nanostructures at different position is very small, indicating that the nanostructures is uniform at different positions among the entire surface.

The experimental repeatability of these nanostructures has been demonstrated, by using a tracking sample for each run of RIE treatment and is illustrated in Figure S2 in this response letter. As shown in Figure S2, for nanostructures generated in multiple runs, the periodicity is peaking at ~350 nm. Only very small deviation of the average depth can be noted, as shown in Figure S2b. Therefore, the method we presented in this study to generate nanostructures is controllable with good repeatability. Now we add these data into the supplementary information as Figure S2.

Additional revision, **not based on the reviewers' comments**:

In the original version of the manuscript, the term 'extraction efficiency' was used to quantify the ability of the nanostructures to couple out trapped photons. In an independent work of our group, we have recently established a formalism to characterize the outcoupling efficiency in connection with the respective layer architecture of the OLEDs. This formalism takes possible optical limitations of specific devices into account. We have originally called the corresponding quantity 'extraction efficiency' as it was used here, but we have been convinced during the peer review process of this work, that the use of 'extraction efficiency' would cause ambiguities. Ultimately, we have proposed a new term '**efficiency of light outcoupling structures**', **short ELOS**, in the corresponding, recent publication (**Efficiency of Light Outcoupling Structures in Organic Light-Emitting Diodes: 2D TiO₂ Array as a Model System** - <https://doi.org/10.1002/adfm.201901748>) to represent the efficiency that is the result of the above formalism.

To follow this reasoning and avoid ambiguities, we have changed in this current revision from the term 'extraction efficiency' to 'efficiency of light outcoupling structures'. The procedure to calculate it and the reported results remain unchanged.

References:

1. Li, Y., Shi, T., Gao, X. & Tu, G. The fabrication of nanostructures with a large range of dimensions and the potential application for light outcoupling in organic light-emitting diodes. *J. Micromechanics Microengineering* **58**, 625–633 (2019).
2. PlasmaPro 80 RIE - Plasma Technology - Oxford Instruments. Available at: <https://plasma.oxinst.com/products/rie/80rie>. (Accessed: 6th April 2019)
3. Sun, Y. & Forrest, S. R. Enhanced light out-coupling of organic light-emitting devices using embedded low-index grids. *Nat Phot.* **2**, 483–487 (2008).

4. Ou, Q.-D. *et al.* Extremely Efficient White Organic Light-Emitting Diodes for General Lighting. *Adv. Funct. Mater.* **24**, 7249–7256 (2014).
5. Jeon, S. *et al.* High-Quality White OLEDs with Comparable Efficiencies to LEDs. *Adv. Opt. Mater.* **17013491**, 1–8 (2018).
6. Reineke, S. *et al.* White organic light-emitting diodes with fluorescent tube efficiency. *Nature* **459**, 234–238 (2009).
7. Chang, H. W. *et al.* Nano-particle based scattering layers for optical efficiency enhancement of organic light-emitting diodes and organic solar cells. *J. Appl. Phys.* **113**, (2013).
8. Qu, Y., Kim, J., Coburn, C. & Forrest, S. R. Efficient, Non-Intrusive Outcoupling in Organic Light Emitting Devices Using Embedded Microlens Arrays. *ACS Photonics* **5**, 2453–2458 (2018).
9. Zhou, L. *et al.* Efficiently Releasing the Trapped Energy Flow in White Organic Light-Emitting Diodes with Multifunctional Nanofunnel Arrays. **25**, 2660–2668 (2015).
10. Kim, Y. H. *et al.* We Want Our Photons Back: Simple Nanostructures for White Organic Light-Emitting Diode Outcoupling. *Adv. Funct. Mater.* **24**, 2553–2559 (2014).
11. NOA63. Available at: https://www.norlandprod.com/adhesives/NOA_63.html. (Accessed: 6th April 2019)

REVIEWERS' COMMENTS:

Reviewer #1 (Remarks to the Author):

It seems that the authors have addressed the comments of all the reviewers carefully. Now I think it can be published as is.

Reviewer #2 (Remarks to the Author):

The authors have addressed most of the reviewers' comments so that the manuscript can be accepted for publication.

Reviewer #3 (Remarks to the Author):

The authors revised fully the manuscript according to the reviewers' comments, which satisfy the republication requirement.